# Acute Promyelocytic Leukemia (APL): A Review of the Classic and Emerging Target Therapies towards Molecular Heterogeneity

**Tâmara Dauare de Almeida, Fernanda Cristina Gontijo Evangelista and Adriano de Paula Sabino ***

Clinical and Toxicological Analysis Department, College of Pharmacy, Federal University of Minas Gerais, Avenue Presidente Antônio Carlos, 6627 Pampulha, Belo Horizonte 31270-901, MG, Brazil
* Correspondence: adriansabin@ufmg.br

**Abstract:** The occurrence of severe bleeding syndrome because of the PML-RARα fusion protein is a life-threatening event in APL. This protein destabilizes homeostasis, maturation, remodeling, and tissue regeneration in addition to hampering the maintenance and differentiation of hematopoietic cells into different lineages, fixing cells in the promyelocyte stage. APL is a classic example of how effective targeted therapy is and, therefore, how important the use of such therapy is to the overall survival of patients, which in this case is represented by the use of ATRA/ATO. Despite that, about 10% of cases of APL patients demonstrate resistance to treatment. Facing this scenario, we point out promising target therapies such as those recommended by the NCCN and Leukemia Net. Since this is such a heterogeneous molecular disease, it is of great importance to understand how important combined chemotherapy, target therapy, immune-based therapy, and combined therapies are in the survival of these APL patients.

**Keywords:** acute promyelocytic leukemia; target therapy; genetic heterogeneity



## 1. Introduction

Heterogeneity in the universe of leukemias is a constant reality. When it comes to acute leukemias, APL is quite distinct because it has a pathognomonic marker, i.e., the translocation between chromosomes 15 and 17, t(15;17)(q22;q21), which results in the PML-RARα fusion protein (promyelocytic leukemia protein—α-retinoic acid receptor) (Figure 1). The ultimate clinical consequence of this genetic translocation is severe hemorrhagic syndrome, which explains why this leukemia entity is fatal if left untreated [1–3].

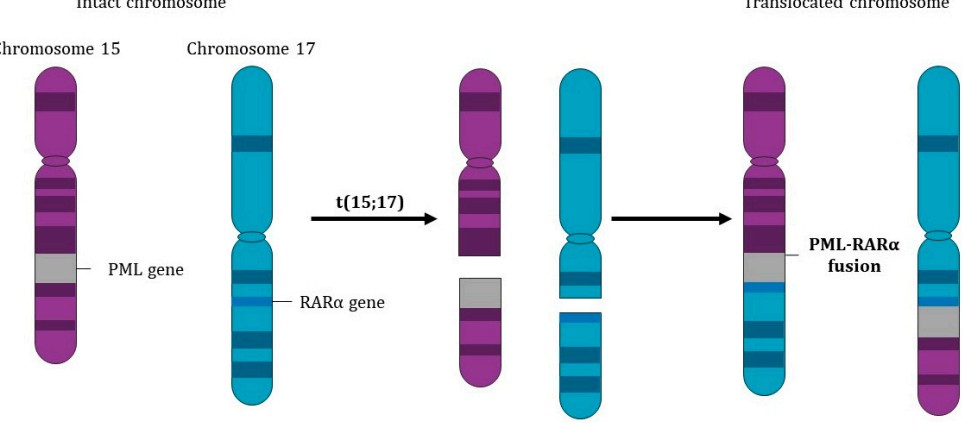

**Figure 1.** Chromosomal translocation between chromosomes 15 and 17. Acute promyelocytic leukemia is associated with the chromosomal translocation t(15;17)(q22;q21), which results in the fusion of genes PML and RARα.

It is classified in the classic FAB system as subtype AML-M3. It is characterized by an infiltration of the bone marrow by leukemic blasts such as promyelocytes. Abnormal promyelocytes have an eccentric nucleus and abundant granulations in the cytoplasm. As for the translocation between chromosomes 15 and 17 that leads to this leukemic manifestation, the PML-RARα fusion protein blocks cell differentiation by interfering with the signaling pathway involving the p53 protein, classically known in the progression of the stages of the cell cycle. The observed result is what is observed in the peripheral blood of these patients under a microscope, namely the accumulation of immature cells trapped in the promyelocyte stage, including in the bone marrow [4].

Epidemiological studies show that the age profile of patients indicates a prevalence of 7% in children and 15% in young adults for manifestations of "de novo acute myeloid leukemia", whereas in adults this manifestation is rare after the age of 45 years old. Some studies point out that, in addition to age, there is an ethnicity factor in the predominance of the disease, since the Latino population has a large proportion of patients with APL among diagnoses of AML [5–7].

APL is the best known example of how targeted therapy is important for patient survival. Since the introduction of all-trans retinoic acid (ATRA) and arsenic trioxide (ATO) into the clinical setting, which revolutionized treatment, the disease-free survival rate has increased to 80–90% [8].

Nowadays, as there is more understanding of possible target therapies, there are more combinations that have the potential to mitigate ATRA and ATO resistance in APL cells. We sought through this work to summarize the studies addressing this need for updated and converging information in relation to targeted therapy use in the treatment of APL.

*1.1. Pathogenesis*

1.1.1. Leading Cause of Death: Disseminated Intravascular Coagulation

Disseminated intravascular coagulation is a condition where an increase in blood coagulation events depletes the platelets and clotting factors needed to control bleeding, thus causing hemostatic challenge and subsequent excessive bleeding. This situation can lead to death in patients with APL. This occurs because promyelocytes can present two pro-coagulant biomarkers: the tissue factor (TF) and the procoagulant cancerous molecule (PC). PCs are released by APL blasts as an alternative procoagulant factor and directly activate factor X of the coagulation cascade [8]. Therewithal, promyelocytes carry inflammatory cytokines, and inflammation has been described as having procoagulant characteristics. Most of all, during treatment, when the cells are entering apoptosis they exteriorize their phospholipid membranes, activating TF. TF, in turn, activates coagulation factor VII and subsequently factor X, whereas PCs can directly activate factor X without the participation of factor VII [9]. In addition to hypercoagulation, the patient may concomitantly develop hyperfibrinolysis by increasing annexin A2 (plasminogen and tPA surface receptor which converts plasminogen into plasmin) or through low levels of fibrinogen. Inflammatory cytokines such as Interleukin-1 (IL-1) and tumor necrosis factor-alpha (TNF-α) modulate TF expression and, in addition, cause endothelial damage that decreases thrombomodulin and elevates the levels of plasminogen activator inhibitor type I (PAI-1) [10].

The main organs affected by hemorrhages in these patients are the brain and lungs, though hemorrhagic manifestations can also occur in the gastrointestinal mucosa and usually have a fatal outcome [11].

1.1.2. The Pathognomonic Genetic Event

Nuclear bodies (NBs) are nuclear substructures found in the cell nucleus and do not have a membrane. They are multiprotein structures that interact with regulatory signals such as nuclear chromatin dynamics and processes such as RNA splicing, gene transcription, and epigenetics, among others. The PML protein occurs within these discrete NBs, and its fusion with the α-retinoic acid receptor due to t(15,17) defines the cellular landscape of APL by repressing the transcription of RARα target genes as a result of

the presence of the PML-RARα fusion protein. Therefore, causing NB disruption affects structures such as RAR, which are in charge of controlling the homeostasis of tissue growth, shaping, and regeneration. However, ligand-free RARα binds to response elements in DNA repressing gene transcription. The upshot is the abruption of hematopoietic stem cell development, maintenance, and expansion, as well as maturation/differentiation in distinct hematopoietic cell lineages. Specifically, what the binding of ATRA causes is a conformational change that results in corepressors dissociating and activating the transcriptional mechanism of genes in this region (Figure 2) [12].

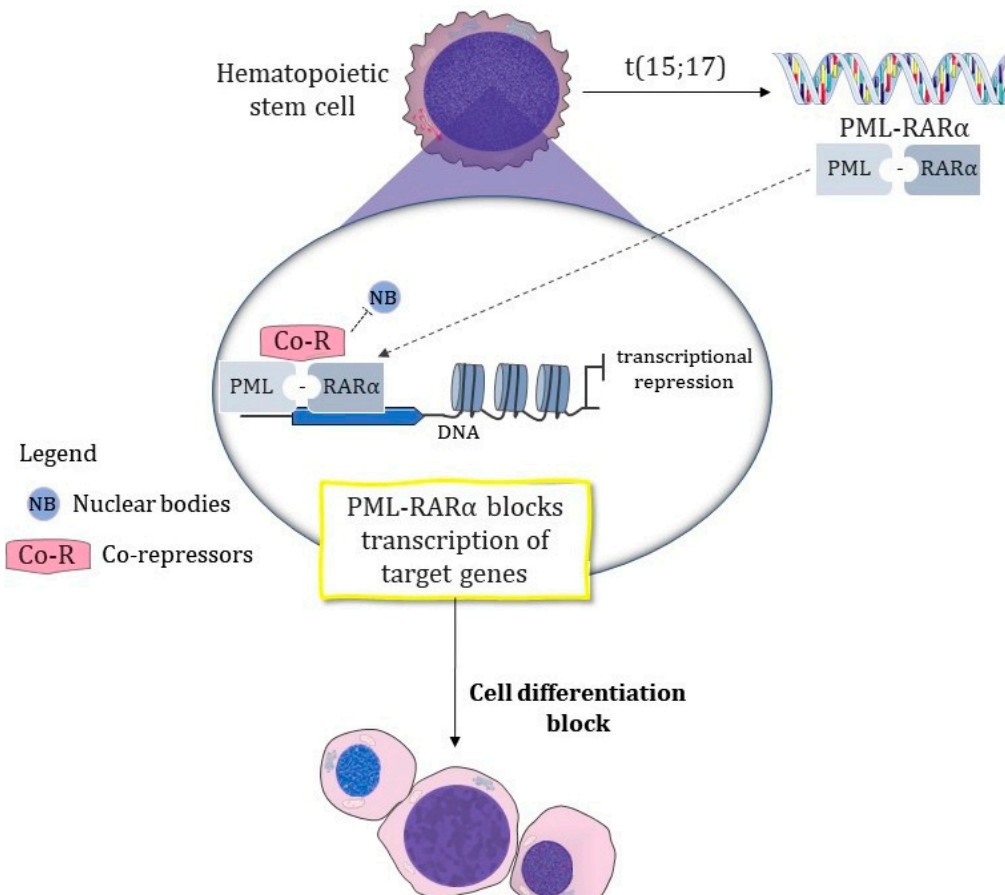

**Figure 2.** RAR transcriptional pathway regulating hematopoietic development through PML-NB disruption. The presence of the PML-RARα fusion protein causes the repression of a transcriptional complex of RAR genes, compromising hematopoietic cell maturation.

Senescence, for example, has its signaling pathway related to this gene region. Many cellular processes act as defense mechanisms that prevent cell transformation from perpetuating, such as programmed cell death through senescence. The PML protein in NBs is sensitive to this cellular stress signaling and is recruited by the p53 signaling pathway; however, cells with compromised PML function demonstrate resistance to the processes of senescence and apoptosis, even during a p53 activation event [13].

The PML-RARα chimeric protein maintains blockages of RAR binding domains and is thus considered the critical oncogenic event that leads to APL pathology. Two main mechanisms are proposed: promoting resistance to apoptosis and disrupting the transcription of RAR targets such as epigenetic cell differentiation. The p53 protein has already had its importance in the cell cycle very well characterized. Fundamentally, upon cell damage signaling, p53 through other proteins causes cell cycle arrest during the mitosis G1 phase of cell repair before the S phase and cell division. The transcriptional repression is indirect and requires p21 to mediate this cycle arrest in order to interact with the cyclin kinase

pathway. The activated p21 can phosphorylate retinoblastoma-associated protein (Rb), which in turn is a regulator behaving as a corepressor through E2F transcriptional factor interaction. E2F acts as a transcriptional factor, binding to DNA regions of gene promoters such as Cyclin A. The Cyclin A protein performs a crucial role during the S phase, but also helps during $G_2/M$ transition. This would explain part of the association of p53 interfering during the G2/M phase. At this stage, DNA replication is already completed and the cells are ready for cell division; however, if the lack of cell repair is noticed or this positive feedback is overwhelmed, then it can lead to apoptosis/senescence for other protein interactions such as those involving the BCL-2 family and regulators of this intrinsic apoptosis pathway. This mechanism elucidates the notion of genome instabilities being key events for tumorigenesis through interference with the p53 mechanism of action, either by a mutation in the protein itself or in the complexes it recruits, such as NBs containing PML. Concerning APL, the PML-RARα complex interacts with the p53 mechanism and has dominant negative action toward transcription, thus inhibiting activation by physiological ligands and causing maturation arrest at the promyelocyte stage (Figure 3) [6,13,14].

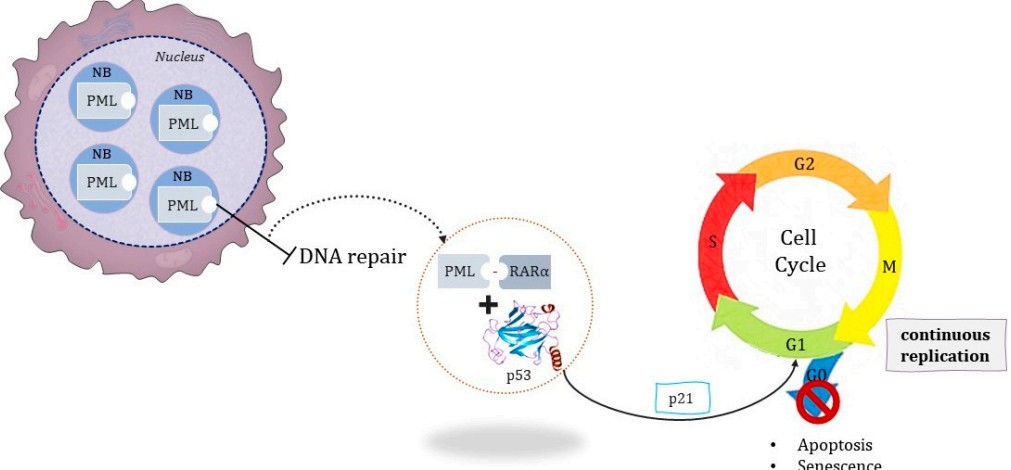

**Figure 3.** PML-RARα interaction with p53 interrupting DNA repair and apoptosis. PML proteins are generated in NBs and are related to cell maturation and senescence through the p53 signaling pathway. Therefore, the association of chimeric protein PML-RARα to p53 protein leads cells to continuously replicate and become non-responsive to cellular signals of senescence and apoptosis.

Despite comprehension of PML-RARα actuation, APL seems to be caused by a range of genetic factors, not just a single mutation event. An example of this is the development of APL driven by PML-RARα in murine tumors requiring secondary oncogenic events, such as *Wilms' tumor 1* (*WT1*), *KRAS*, *NRAS* mutations, and *FMS-like tyrosine kinase 3* (*FLT3*) activation [15].

### 1.1.3. Associated Mutations

*FLT3* is a gene that belongs to the class III receptor tyrosine kinase (RTK) family. RTKs are well correlated with cell proliferation, and *FLT3* mutations are recurrently associated with AML prognosis [16].

Under physiological conditions, transcription of the *FLT3* gene encodes a monomeric protein consisting of an extracellular domain, a transmembrane portion, a juxtamembrane (JM) domain, and two intracellular tyrosine kinase downstream signaling proteins. In the presence of ligands, dimerization of monomers occurs, followed by the phosphorylation of effector substrates of intracellular signal transduction pathways. When the *FLT3* gene is mutated, it gives rise to a modified final product, that is, it generates an *FLT3* receptor with changes in its structure [17].

Two types of *FLT3* activation mutations have been identified in AMLs. Internal tandem duplications (ITDs) affect the juxtamembrane (JM) either in tyrosine kinase domain

1 (TKD1) of the *FLT3* receptor or at the predefined point mutation in tyrosine kinase domain 2 (TKD2). As a result, the tyrosine kinase domain is permanently activated, regardless of ligand, which leads to the uncontrolled proliferation of myeloid cells. This deregulated activation impairs hematopoiesis and will contribute to leukemogenesis. *FLT3* mutations are present in approximately 2% to 38% of APL cases, depending on ITDs or mutations in the tyrosine kinase domain [17–19].

Since 2008, after a publication from the International Agency for Research on Cancer (IARC) included a molecular parameter for leukemia classifications and stratification, *FLT3* mutations have been identified as being highly related to hyper leukocytosis. Regarding APL, *FLT3-ITD* impedes ATRA-induced differentiation. Such resistance was abrogated when combined with arsenic trioxide. This is a strong model that indicates the key role of arsenic in targeting PML/RARα for destruction through distinct biochemical pathways related to ATRA [20].

*Wilms' tumor* 1 (*WT1*) was first identified as a predisposition gene for Wilms' tumor, functioning as a tumor suppressor. In hematologic neoplasms, *WT1* is targeted by somatic mutations in 6% to 15% of "de novo AML cases". A concurrent occurrence of *WT1* with *FLT3-ITD* has been identified in pediatric AML, leading to worse overall survival than the presence of either mutation alone [21].

In hematopoietic progenitor cells presenting CD34+, *WT1* is present at low levels; however, for AML, and especially in the APL scenario, it is highly expressed. This fact promotes its expression profile as a relapse marker in APL [22].

The RAS protein is an extracellular signal transducer and an important pathway in the transmission of information from the cell membrane to the nucleus. RAS proteins play a key role in regulating many cellular events, including cell proliferation, migration, survival, and apoptosis. Mutant RAS proteins translate these downstream signals and, consequently, have an oncogenic role. The RAS mutations known as *HRAS*, *NRAS*, and *KRAS* are among the most common oncogenes, and about 19% of cancer patients have an RAS mutation. Patients with mutant RAS have a worse prognosis and shorter overall survival. *NRAS* mutations are present in 8–13% of AML patients, while *KRAS* mutations can be found in 2% of these patients [23,24].

*FLT3* and *KRAS* have direct implications on senescence driven through PML action. Cell death caused by RAS-activated senescence is lost in circumstances where there is an absence or mutation of the PML protein. Interestingly, *FLT3*- and *KRAS*-deregulated expression is described as a cooperator of PML-RARα in mouse models of APL [25,26].

## 2. Target Therapy Has Been the Mainstay Treatment for APL since the 1980s

APL treatment has undergone important modifications over the last 30 years and differs from the regimens used for other AMLs. The greatest impact on the treatment of APL was undoubtedly the demonstration that ATRA, in pharmacological doses, allows the progression of cell differentiation. Thus, the leukemic clone progresses in myeloid maturation, becoming susceptible to cell death mechanisms and chemotherapy. Notably, two of the most active drugs in APL therapy (ATRA and ATO) allow for the reformation of PML NBs because of the degradation of PML-RARα. ATO has a complex mechanism of action and is believed to interfere with multiple intracellular signaling pathways by binding cysteine-rich residues. Directly, ATO leads PML-RARα toward ubiquitination and proteasomal degradation. This therapeutic conduct resulted in complete remission in more than 90% of cases and a 5-year disease-free survival rate of approximately 80% [6,10,27,28].

ATRA's most common side effects are similar to those seen in a person who takes a lot of vitamin A. These symptoms can include headache, fever, dry skin and mouth, rash, feet swelling, sores in the mouth or throat, itching, and eye irritation. For patients with APL, great attention is needed to notice symptoms of differentiation syndrome (DS) (also known as ATRA syndrome) that are manifested with cerebral, pulmonary, and myocardial involvement. Generally, decreasing the dose and use of the DS protocol decreases toxicity without affecting treatment efficiency [29].

The pioneering concept of ATRA's mechanism of action relates to its binding to the receptor directing cells toward differentiation, thus removing them from the promyelocyte stage, while ATO activates cascades of apoptosis or partial differentiation of cells. Elucidations in animal models and bone marrow examinations of patients with APL demonstrated that high concentrations of ATO induced apoptosis by activating the mechanisms of apoptosis via mitochondria. On the other hand, at low concentrations, ATO promoted differentiation. Over the years, the pro-apoptotic activity of ATO has been investigated at the molecular level and its impact on genes and proteins has been described. Critically, arsenic does not activate RAR-dependent transcription, although it can induce differentiation syndrome in APL patients [30]. Subsequent research has shown that the therapeutic efficacy of ATO is due to its effect on the PML portion of the PML-RARα fusion protein (Figure 4). ATO binding to the PML component via two cysteine residues induces its oxidation and the formation of a disulfide bond. In the final step, ATO mediates the recruitment of the 11S proteasome, a process essential for the degradation of PML and PML-RARα proteins. PML-RARα destruction also abrogates self-renewal through the reformation of PML nuclear bodies that were initially disassembled by PML-RARα and the subsequent activation of p53 [8,28,31].

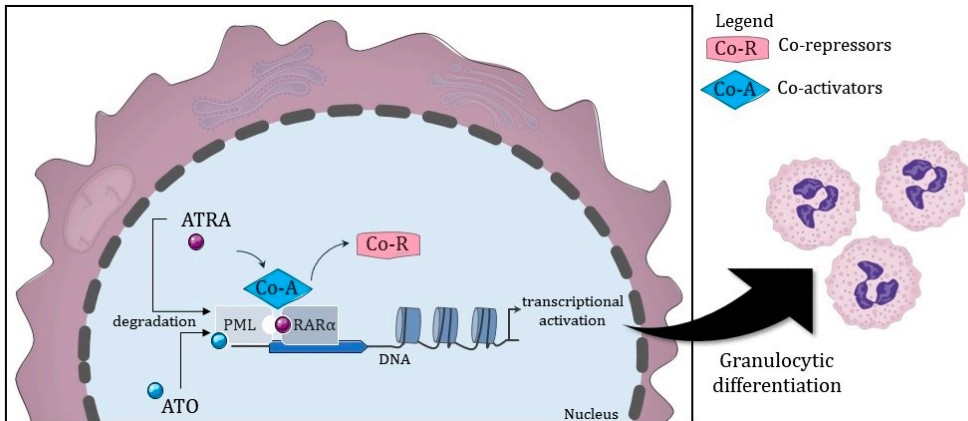

**Figure 4.** ATRA and ATO binding and dissociating corepressors leading to cell differentiation. APL target therapy deactivated the repression complex caused by the PML-RARα protein and returned genetic transcription to cell differentiation, which leads cells back to responsiveness to the cell death mechanisms of chemotherapy.

The National Comprehensive Cancer Network (NCCN) protocol currently recommends the more recently released drugs when it comes to target therapies for AML. Targeted therapy is a drug therapy that focuses on the specific or unique characteristics of cancer cells. Targeted therapy can be used alone or in combination with chemotherapy. Its major advantages are specificity and reduced side effects. Examples of drugs that have already been approved include Gemtuzumab, Midostaurin, Gilteritinib, Sorafenib, Venetoclax, Enasidenib, and Ivosidenib [32].

Gemtuzumab is a conjugated drug, specifically, it is a monoclonal antibody directed to CD33 that is covalently linked to the cytotoxic agent N-acetyl-γ-calicheamicin (ozogamicin). Its efficacy is rooted in the ubiquitous nature of CD33 as an antigen marker in AML patients since it is present in blasts from 90% of AML patients. This makes it a potential target marker of an immunotherapeutic option for AML [33] (Figure 5). Clinical observations demonstrate the safety and efficacy of using a strategy of ATRA, ATO, and Gentuzumab-Ozogaminicin (GO) administration, with a benefit of this approach being that it is a better tolerated therapeutic regimen in high-risk patients than the combination of ATRA and chemotherapy [34].

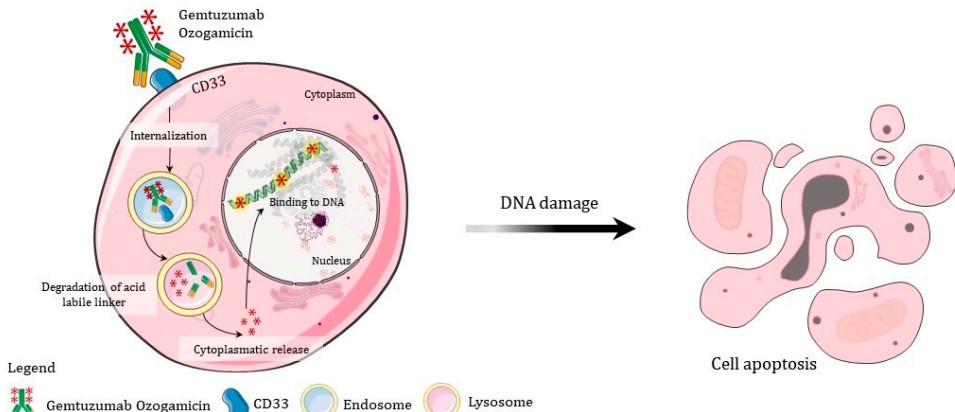

**Figure 5.** CD33-targeting Gemtuzumab-Ozogamicin. Once the CD33 receptor drives the molecules to endocytosis and degradation, the cytotoxic agent causes DNA damage and promotes cell death.

*FLT3* is a tyrosine kinase receptor and an important early regulator of hematopoiesis whose mutations affect prognosis in AML patients. Internal tandem duplication mutations of *FLT3 (ITD)* result in uninterrupted activation of pathways and their signaling cascades that promotes cell survival and proliferation, including the *MAPK/ERK*, *PI3K/AKT*, and *JAK/STAT* pathways. Direct inhibition of *FLT3* is therefore a promising therapeutic route, with some agents already available for immediate use. The US Food and Drug Administration (FDA) approved Midostaurin (a first-generation multikinase inhibitor) in 2017 for use in the treatment of AML patients with *FLT3* mutations. This protocol also involved its combination with cytarabine and induction with daunorubicin and cytarabine consolidation. However, it is known for patients over 60, mostly those with other comorbidities, that the chemotherapy protocol has extensive toxicities and significant effects. Meanwhile, in 2018, Gilteritinib was approved for clinical use in the treatment of adult AML patients with poor responsiveness due to *FLT3* mutations (Figure 6). Sorafenib, previously approved, is another *FLT3* kinase inhibitor worth mentioning. Studies have shown that Sorafenib can inhibit cell proliferation by inducing cycle arrest and causing apoptosis in APL cells, probably by interfering with the *MEK/ERK* signaling pathway. These multiple kinase inhibitors target not only *FLT3*, but also other kinases, revealing the antileukemic effects of these non-specific inhibitors [35,36].

Venetoclax is an oral BCL-2 inhibitor. BCL-2 mediates tumor cell survival and is associated with chemotherapy resistance. Venetoclax binds directly to BCL-2 protein and selectively inhibits BCL-2 by relocalizing pro-apoptotic proteins and restoring apoptosis (Figure 7) [37]. Venetoclax is particularly capable of crossing the blood–brain barrier and could be an option for treating patients suffering from APL relapse in the central nervous system [38].

Up to 19% of AML patients have mutations in isocitrate dehydrogenase-2 (IDH2). IDH2 mutations produce an oncometabolite, 2-hydroxyglutarate (2-HG), which leads to hypermethylation of DNA and histones and impaired hematopoietic differentiation. Enasidenib is an oral inhibitor of mutant IDH2 proteins and was well tolerated by AML patients. A previous study by Stein et al. (2017) reported hematologic remission responses in refractory AML patients [39].

Isocitrate dehydrogenase 1 (IDH1) is a metabolic enzyme that catalyzes the oxidative decarboxylation of isocitrate to a-ketoglutarate (a-KG). Mutations in IDH1 occur in up to 10% of patients with AML. Mutant IDH1 (mIDH1) catalyzes the reduction of a-KG to the oncometabolite 2-HG. 2-HG competitively inhibits a-KG-dependent enzymes, leading to epigenetic changes and impaired hematopoietic differentiation. Ivosidenib is an oral small molecule targeted mIDH1 inhibitor. Ivosidenib monotherapy has been well tolerated and induces durable remission. Ivosidenib and Enasidenib are important in the context of APL as mutations in IDH1/2 demonstrate the idea that APL and AMLs carrying mutated IDH carry common pathways and therefore treatments (Figure 8) [40].

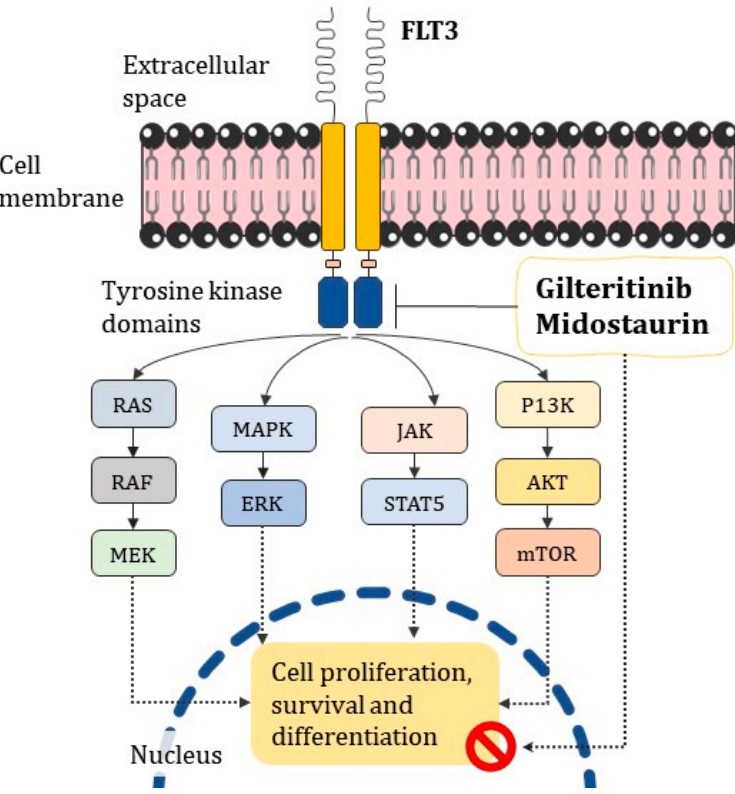

**Figure 6.** Gilteritinib and Midostaurin mechanism through *FLT3* kinase inhibition. *FLT3* mutations lead cells to proliferation, survival, and a lack of differentiation. Gilteritinib, Midostaurin, and Sorafenib are intracellular ligands to kinase receptors that block this cascade effect, returning metabolism to cell division and cell death control.

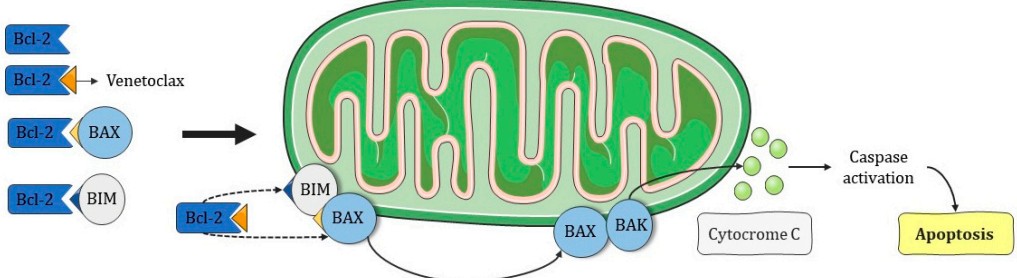

**Figure 7.** Venetoclax mechanism through BCL-2 inhibition. BCL-2 associated protein X (BAX) is a pro-apoptotic protein that is recruited by Venetoclax. In the presence of an apoptotic signal, BAX is ranslocated from the cytoplasm to the vicinity of the mitochondria, where it undergoes activation and conformational modification before adhering to the outer mitochondrial membrane. Small units of activated BAX proteins form oligomers that eventually penetrate the outer mitochondrial membrane and release cytochrome c, which activates the cell cascade to apoptosis via caspases (adapted from Kucukyurt and Eskazan, [37]).

*2.1. Recommended Treatment*

Upon diagnosis, APL patients should be classified as low risk when their global white blood cell (WBC) count is fewer than $10 \times 10^9$/L or high risk when their WBC count is greater than $10 \times 10^9$/L (NCCN recommendations). The recommendations from an expert panel of the European Leukemia Net (ELN) made in 2019 encourage the immediate administration of ATRA in patients suspected of suffering from APL. They also determined that an increase in WBC count above $10 \times 10^9$/L after treatment with ATRA or ATO should be interpreted as a sign of ATRA/ATO-induced differentiation and that the risk should not

be reclassified. For these cases, the administration of hydroxyurea, idarubicin, or GO is recommended in case of hyperleukocytosis. Coagulation homeostasis must be monitored daily and controlled; therefore, transfusions of fibrinogen, platelets, and fresh frozen plasma should be used to maintain fibrinogen concentration above 100–150 mg/dL, platelet count between $30 \times 10^9$/L to $50 \times 10^9$/L, and the INR (International Normalized Ratio) at a value below 1.5 [41].

The treatment of AMLs, and therefore APL, is normally divided into two phases: induction and consolidation (post-remission therapy). Induction therapy is used as the first step in efforts to reduce the spread of cancer and is also used in the evaluation of the drug response. Patients with different risk classifications will have different induction protocols.

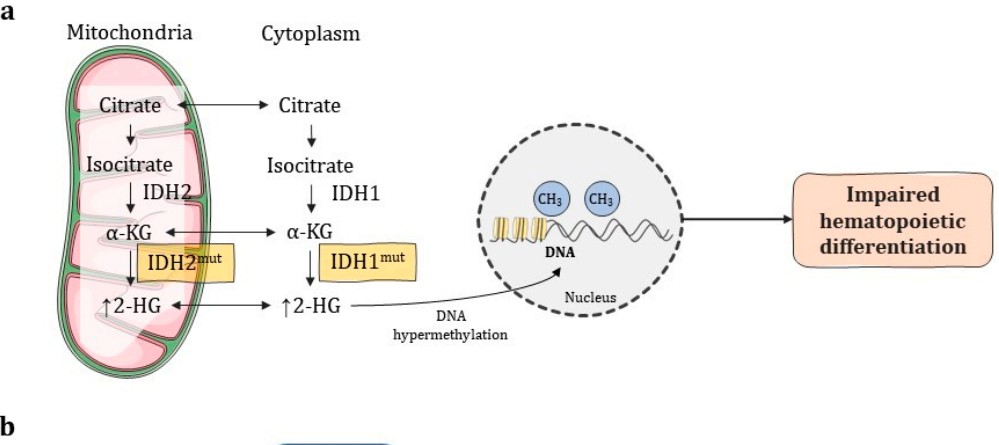

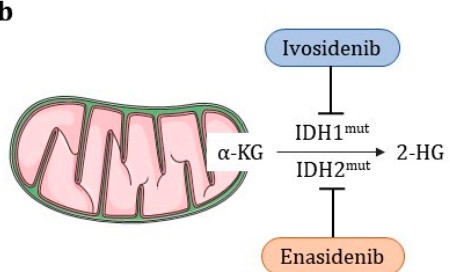

**Figure 8.** Ivosidenib and Enasidenib mechanism through mitochondrial citrate metabolism. Mutations in IDH1 and IDH2 enzymes drive accumulation of the 2-HG citrate metabolite, causing DNA hypermethylation and impaired hematopoietic differentiation (**a**). Enasidenib and Ivosidenib act on mutated IDH1mut and IDH2mut, restoring 2-HG citrate metabolite levels and returning cell metabolism to differentiation and sensitiveness to cell death (**b**).

Recommendations for low-risk cases are summarized in Table 1 below as options A or B.

**Table 1.** Recommended therapy for low-risk APL patients.

| | DRUGS | DOSE | MOMENT | PERIOD |
|---|---|---|---|---|
| A | ATRA<br>ATO | 45 mg/m$^2$ daily (in 2 divided doses)<br>0.15 mg/kg IV I | Daily until remission | |
| B | ATRA<br>ATO<br>ATO | 45 mg/m$^2$ daily (in 2 divided doses)<br>0.3 mg/kg IV I<br>0.25 mg/kg | Daily until remission<br>Days 1,2,3,4,5<br>Twice weekly | <br>Week 1<br>Weeks 2–8 |

IV I: Intravenous infusion; [32].

Children and adolescents can be administered a 25 mg/m$^2$ dose of ATRA. In cases where ATO must not be used, idarubicin at a dose of 12 mg/m$^2$ can be used, or alternatively a single 9 mg/m$^2$ dose of GO on day 5.

For high-risk patients who tolerate anthracyclines, the NCCN recommends divided doses of ATRA (45 mg/m$^2$) until clinical remission and one of the following regimens in Table 2 below.

**Table 2.** Recommended therapy for high-risk APL patients.

| | DRUGS | DOSE | MOMENT | PERIOD |
|---|---|---|---|---|
| A | ATRA + IDARUBICIN ATO | 6–12 mg/m$^2$ 0.15 mg/kg | Days 2, 4, 6, 8 Days 09–36 | 2 h IV I |
| B | ATRA + ATO SINGLE DOSE OF GO | 0.15 mg/kg/d IV I 9 mg/m$^2$ | Day 1, 2, 3, or 4 | |
| C | ATRA + ATO ATRA + ATO SINGLE DOSE OF GO | 0.3 mg/kg IV I 0.25 mg/kg 6 mg/m$^2$ | Days 1, 2, 3, 4, 5 Twice weekly Day 1, 2, 3, or 4 | Week 1 Weeks 2–8 |
| D | ATRA + DAUNORUBICIN CYTARABINE | 50 mg/m$^2$ IV I 200 mg/m$^2$ IV I | Days 3, 4, 5, 6 Days 3, 4, 5, 6, 7, 8, 9 | |
| E | ATRA + DAUNORUBICIN CYTARABINE | 60 mg/m$^2$ 200 mg/m$^2$ IV I | | 3 days 7 days |
| F | ATRA+ IDARUBICIN | 12 mg/m$^2$ | Days 2, 4, 6, 8 | |

IV I: Intravenous infusion [32].

For high-risk patients unable to tolerate anthracyclines, the NCCN recommends ATRA plus ATO. Induction must be continued until bone marrow recovery and remission [32].

*2.2. Consolidation Therapy*

According to NCCN recommendations, the agents used for induction therapy should be sustained during consolidation therapy, while the ELN recommends that high-risk patients should also receive chemotherapy and low-risk patients should receive antileukemic agents after genetic diagnosis of APL (for antileukemic agents, we refer to ATRA/ATO treatment). However, after consolidation, the ELN does not recommend maintenance therapy and suggests not performing routine monitoring for low-risk patients with negative minimal residual disease (MRD); however, for high-risk patients, they suggest that health centers should monitor MRD quarterly for three years [42].

**3. Differentiation Syndrome and Its Problems**

Patients with elevated WBC counts are more likely to relapse and are more prone to develop differentiation syndrome (DS) [30]. This syndrome has a very high incidence in APL patients treated with ATRA, usually occurring within 2 to 21 days of the initiation of treatment. DS may manifest with unexplained fever, hypotension, respiratory distress, pulmonary infiltrates, pericardial effusions, and renal impairment. The pathogenesis of DS has not been well defined. The hypothesis derived from research in primary cultures and cell lines suggests a systemic inflammatory response driven by the release of pro-inflammatory cytokines, with endothelial damage, capillary leakage, and alterations in the expression of cell adhesion molecules leading to tissue infiltration by leukemic cells and occlusion of the microcirculation [43].

Some studies have pointed to an increase in mature WBCs susceptible to the exacerbated inflammatory process. Therefore, ATRA would increase the expression levels of cell adhesion molecules and promote an excessive systemic inflammatory response, which would ultimately lead to endothelial cell damage, capillary permeability, vessel occlusion, and massive tissue infiltration of differentiating APL cells. Nevertheless, the consensus in severe cases of DS (respiratory or acute renal failure) is that discontinuation of ATRA and/or ATO treatment is mandatory [15,19].

Once DS has been recognized, the accepted standard management is treatment with corticosteroids such as dexamethasone, with the suggested dose being 5–10 mg taken twice

daily intravenously until resolution and for at least 3 days after. The use of prophylactic steroid therapy during induction shows a beneficial reduction in the incidence of DS [41].

## 4. Other Mutations That Evade Conventional Treatment

Besides the classic mutations in acute leukemias, particular cases in APL have been investigated, especially regarding responsiveness to available treatments.

Data suggest, for example, that *FLT3*-ITD in acute promyelocytic leukemias prevents ATRA responsiveness, but not responsiveness to ATO [44,45].

Just as there is translocation shaping the PML-RARα protein, translocation can link other genes whose prognoses become even more heterogeneous. Other translocational rearrangements of the *RARA* gene have also been reported [43].

However, reported variants with *NPM1*, *FIP1L1*, *NUMA1*, *IRF2BP2*, *GTF2I*, *IRF2BP2*, *FNDC3B*, *BCoR*, *NPM*, *OBFC2A*, and *PRKAR1A* were shown to be sensitive to ATRA; therefore, chemotherapy and target therapy together could benefit this cohort. Other variants with *ZBTB16* and *STAT5b* responded weakly, while the *TBLR1* variant was not responsive to ATRA treatment [46].

## 5. Possible Treatments for Each Mutation

APL treatment since the 1980s has been studied as a different entity, which coincides with the change in status from leukemia being seen as highly lethal to a disease that sees remission in about 90% of cases. Nevertheless, for this 10% of new cases of APL, scientific research has been investigating its causes and seeking better therapeutic management. Besides the classic cases of translocation that do not respond well to the classic therapy for APL, other hypotheses exist for this evasion of therapy [47]:

- The use of cytotoxic drugs, especially topoisomerase inhibitors, alkalinizing agents, and anthracyclines, leads to changes in DNA structure and secondary changes, most of which include treatment-related myelodysplastic syndrome or AML;
- APL is accompanied by other clones in the early stages of the disease but masked by the dominant clones because the application of ATRA, ATO, and chemotherapy eliminates abnormal promyelocytes so that other clones have a better chance of survival;
- As reported in the literature, the prognosis of APL, secondary to treated acute myeloid leukemia, is poor and survival is relatively short. Although patients have achieved complete remission after one cycle of chemotherapy, their long-term survival still needs to be investigated because it is unclear whether the choice of follow-up consolidation therapy should be medium-dose cytarabine-based chemotherapy or allogeneic hematopoietic stem cell transplantation.

In general, resistance cases have been treated with the corresponding conventional AML treatments. However, we need to converge the applicability of the promising target therapies and those recommended by the NCCN:

- Gentuzumab is recommended for immature cells with CD33 markers; thus, if it is not a blastic crisis, perhaps other drugs may be more effective;
- Midostaurin and Gilteritinib targeting *FLT3*, which is commonly reported in oncohematological disorders, may help in cases where it is uncertain whether conventional ATRA/ATO treatment is effective;
- Enadesinib and Ivodesinib have shown good results for cases in which the previous remission was also impaired;
- Venetoclax acts by restoring apoptosis in cells poorly responsive to chemotherapy;
- Since this is such a heterogeneous disease, it is of great importance to outline the appropriate combined therapy of chemotherapy and target therapy to improve survival in APL patients.

## 6. Emerging and Combined Target Therapies

Recently approved by the FDA, Olutasidenib is recommended to adult patients with relapsed or refractory acute myeloid leukemia (AML) with a susceptible IDH1 mutation

at arginine (R)132 (IDH1(R132)). The recommended Olutasidenib dose is 150 mg taken orally twice daily on an empty stomach, and a complete remission (CR) rate of 35% has been observed [48].

On the other hand, 3 + 7 cytarabine plus daunorubicin chemotherapy has been updated for AML patients. Recent studies including alternative combinations with target therapies that have been used to substitute classic chemotherapy have shown interesting disease-free survival rates. Combined target therapy is uncovering potential therapeutic strategies for AML [49]. The combination of target therapy is a current area of interest, as can be seen by FDA approval in May 2022 of the combination of Ivosidenib and the hypomethylation agent Azacitidine (AZA) for the treatment of AML patients with susceptible IDH1 mutations [50]. For newly diagnosed patients, promising results were reported for the combination of AZA, Venetoclax, and Decitabine. For wild-type *FLT3* and high-risk patients, physicians observed beneficial synergism between Venetoclax and Gilteritinib, in which the mechanism seems to be MCL-1 suppression. Regarding *FLT3*-complicated mutations, the use of Gilteritinib plus AZA in older patients raised the response rate. For the common high-risk p53 mutation, a specific drug named Eprenetapopt has been used in combination with AZA. This same approach (Eprenetapopt plus AZA) had exciting results in post-allogenic HCT therapy. The combination of AZA and Ivosidenib or Enasidenib led to the complete remission of IDH1/2 complications [51–57] (see Table 3). Other ongoing studies for APL therapy can be seen in Table 4 below.

**Table 3.** Emerging combined target therapies.

| Indication | Combination |
|---|---|
| IDH1/2 mutation | Ivosidenib +AZA [50] or AZA + Ivosidenib [56] or AZA+ Enasidenib [57] |
| Newly diagnosed, ineligible for intense chemotherapy | AZA+ Venetoclax + Decitabine [51] |
| *FLT3* high-risk | Venetoclax + Gilteritinib [52] |
| *FLT3* mutation in older patients | Gilteritinib + AZA [53] |
| TP53 mutation | Eprenetapopt +AZA [54] |
| Post-allogenic HCT | Eprenetapopt +AZA [55] |

**Table 4.** Ongoing trials of APL therapy.

| Clinical Trial Identifier | Phase | Clinical Test | Related Conditions | Reference |
|---|---|---|---|---|
| NCT02129101 | I/Ib | To evaluate the tolerable dose and efficacy of Azacitidine and Sonidegib or Decitabine | Acute myeloid leukemia Acute promyelocytic leukemia | [58] |
| NCT03625505 | I | To evaluate the safety and efficacy of Venetoclax in combination with Gilteritinib | Relapsed or refractory (R/R) acute myeloid leukemia Acute myeloid leukemia Acute promyelocytic leukemia | [59] |
| NCT03048344 | I | To determine the recommended dose of arsenic trioxide capsule formulation ORH 2014 | Acute myeloid leukemia Acute promyelocytic leukemia Chronic myelomonocytic leukemia Mantle cell lymphoma Myelodysplastic syndromes Myelodysplastic/myeloproliferative neoplasm | [60] |

**Table 4.** *Cont.*

| Clinical Trial Identifier | Phase | Clinical Test | Related Conditions | Reference |
|---|---|---|---|---|
| NCT04655391 | Ib | To evaluate the best dose and effect of Glasdegib with Venetoclax and Decitabine; Gilteritinib, Bosutinib, Ivosidenib, or Enasidenib | Relapse after HCT Acute myeloid leukemia Acute promyelocytic leukemia | [61] |
| NCT03386513 | I/II | To evaluate the antileukemic activity of IMGN632 when administered as a monotherapy to patients with CD123+ | Acute lymphoblastic leukemia Acute myeloid leukemia Acute promyelocytic leukemia Blastic plasmacytoid Dendritic cell neoplasm Chronic myeloid leukemia Chronic myelomonocytic leukemia Myelodysplastic syndromes Myeloproliferative neoplasm | [62] |
| NCT03328078 | I/II | To evaluate oral administration of CA-4948 (reversible inhibitor of Interleukin-1) | Adult patients with relapsed/refractory hematologic malignancies Acute myeloid leukemia Acute promyelocytic leukemia B-cell non-Hodgkin lymphoma | [63] |
| NCT02124174 | II | To evaluate Vidaza and valproic acid post allogeneic transplant | High-risk AML post allogeneic transplant Acute myeloid leukemia Acute myeloid leukemia with t(8;21), (q22; q22.1), RUNX1-RUNX1T1 Acute promyelocytic leukemia Core binding factor Acute myeloid leukemia Myelodysplastic syndromes | [64] |

## 6.1. Immunotherapies

Immune-based approaches become promising once antigens are expressed differently to normal immune cells. Here, we mention the overexpressed antigens in an acute myeloid context for target-directed immune therapy. A canonical example is GO, which is directed to CD33. In the same way, blocking CD47, a "don't eat me signal" which is a macrophage target agent, can enhance anti-tumor activity; however, it has a low clinical response [65]. Additionally, studies on combined drugs and Magrolimab (anti-CD47) are in progress with AZA and even with Atezolizumab, which is another immune checkpoint (NCT03922477) [66]. The CD70 marker promotes blast stemness, which makes it a target for treatment. The anti-CD70 drug Cusatuzumab seems to have an eliminative effect on acute myeloid leukemia cells in preclinical studies, while for clinical approaches, the hypomethylating agent AZA was also combined (NCT03030612) [67]. Talacotuzumab, a representative of anti-CD123, is currently under evaluation in combination with decitabine (NCT024172145) [68].

### 6.1.1. CAR-T

Clinical application of CAR-T therapy (chimeric antigen receptor T cell therapy) has been a clinical challenge for AML since the target antigens applied to this technology are

common in both leukemia cells and normal bone marrow cells. Studies are applying this technology in the context of relapse or refractory AML. Obstacles emerge from different nuances. For example, there have been reports of graft vs. host disease caused by mismatched endogenous and exogenous peptides [69]. In addition, the expression level of these antigens in leukemia cells is often weaker, as can be seen in clinical trial NCT04097301, which was discontinued due to the lower-than-expected proportion of myeloma and leukemia cells expressing the target (CD44v6) [70]. Other studies attempted different strategies. The NCT03473457 clinical trial aimed to determine the toxicity profile of different targets, such as CD38, CD33, CD56, CD123, CD117, CD133, CD34, or MUC1-targeted CAR-T, but the therapeutic effect was not as expected. NCT03126864 concerned autologous T cells transduced with lentivirus that were developed to express a CD33-specific chimeric antigen receptor, though this trial was also terminated. Another strategy proposed in clinical trial NCT03672851 was an anti-CD123 CAR-T treatment involving T cells lentivirally transduced to express a CD123-specific hinge-optimized CD28 costimulatory chimeric antigen receptor; however, this treatment exhibited adverse effects [71–73].

### 6.1.2. Cancer Vaccines

Cancer vaccines have two main approaches: peptide vaccines and dendritic cell vaccines. Regarding leukemias, the immune-compromised system is a barrier that needs to be trespassed in order to achieve antigen recognition. Nevertheless, there are key antigens leading the developmental vaccine race against myeloid leukemias such as WT1. An ongoing phase 2 clinical study (NCT01266083) is trying to determine if the WT1 vaccine causes immune recognition safely enough to prevent leukemia relapse [74]. There are reports of dendritic cell-based immunotherapies for AML, most of which aim to prevent leukemia relapse. Through a major histocompatibility complex, dendritic cells can stimulate adaptative and innate immune responses. These dendritic cell vaccines have the advantage of safety and high immunogenicity. Clinical trials have shown complete response rates ranging from 33% up to 74%. However, specifically for APL, they have no consistent robust data yet [75,76].

## 7. Hematopoietic Cell Transplantation

It is noteworthy that APL is deadly if untreated; therefore, APL treatment stands in a different position to the rest of the AML subclassifications in terms of relapse or refractory cases. If a patient has MRD, and they had been treated classically with chemotherapy plus ATRA, then they should receive a cycle of ATO treatment for reinduction, which has been shown to lead to up to 95% remission. However, in case these become ineffectual, management through a hematopoietic stem cell transplant (HCT) is most recommended [77,78]. To achieve pre-transplant or post-transplant induction, all combinations cited above in this article can apply.

Regarding APL specificity, an HCT is indicated for relapsing patients after non-responsiveness to ATRA and ATO frontline therapy. An autologous HCT is suggested for MRD-free patients and has demonstrated excellent long-term survival. For patients with refractory disease, an allogeneic HCT can be an option after risk management evaluation [79,80].

## 8. Conclusions

Elucidation of the molecular drift that occurs in APL is increasingly guiding treatment toward the reality of pharmacogenomics. Knowing which specific mutation the patient carries and the best drug for each case reduces toxicity and not only increases survival, but also increases the quality of life of patients. We hope that the whole world population will soon have more access to this type of molecular diagnosis, as well as to the target therapies indicated for each situation.

As molecular biology moves forwards in terms of the investigation of molecular pathogeneses, so does pharmacogenomics. Personalization is the future of pharmacology,

especially when seeking to overcome the risk of relapse and unwanted toxicity while improving life expectancy and patient survival rates.

**Author Contributions:** T.D.d.A. collected and organized relevant research, wrote the manuscript, and drew the figures; F.C.G.E. proofread the manuscript and A.d.P.S., T.D.d.A.'s doctorate advisor, guided, supervised, and proofread the manuscript. All authors have read and agreed to the published version of the manuscript.

**Funding:** The authors would like to express their sincere appreciation to the Clinical and Toxicological Analysis Department in the Pharmacy College of the Federal University of Minas Gerais.

**Institutional Review Board Statement:** Not applicable.

**Informed Consent Statement:** Not applicable.

**Data Availability Statement:** Data sharing is not applicable.

**Acknowledgments:** Parts of the Figures 2–8 were drawn by using pictures from Servier Medical Art. Servier Medical Art by Servier is licensed under a Creative Commons Attribution 3.0 Unported License (https://creativecommons.org/licenses/by/3.0/). Our gratitude to CAPES, Brazilian research agency, which sponsored T.D.d.A doctorade's scholarship providing fomentation to this work.

**Conflicts of Interest:** The authors declare no conflict of interest.

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
