# Peer review of "Acute Promyelocytic Leukemia (APL): A Review of the Classic and Emerging Target Therapies towards Molecular Heterogeneity"

_futurepharmacol, doi:10.3390/futurepharmacol3010012_

Round 1

Reviewer 1 Report

I have minimal knowledge of the pharmaceutical par, but I was able to review the article from a molecular biological point of view. 

It is a general observation that the sentences written about P53 are often not inaccurate. 

For example, in line 163: "interrupting mitosis at the G1 phase," see line 140 below. 

Line 164: "Pointedly, the cell cycle at G2/M is checked by p53 once again, since at this stage the DNA replication is already completed and the cells are ready for cell division, but if the lack of cell repair is noticed, then it can lead to apoptosis." 

P53 is not a G2/M cell cycle checkpoint protein. 

The MRN protein complex recognizes DNA lesions such as double-strand breaks (DSBs) and recruits and activates the ATM (ataxia-telangiectasia mutated) and ATR (ATM- and Rad3-Related) kinases. Activated ATM and ATR phosphorylate a variety of substrates, including the downstream kinases Chk1 and Chk2, which regulate the cell cycle and induce apoptosis. Another ATM/ATR downstream protein target is Histone H2AX, which is required for the recruitment and accumulation of DNA repair proteins at DSB damage sites.  

The tumor suppressor transcriptional protein p53, which can promote cell cycle arrest, DNA damage repair, apoptosis, and senescence, is central to the DNA damage response pathway. ATM and Chk2 activate P53, which in turn activates the expression of numerous target genes such as the cell cycle regulator p21, the DNA repair protein Rad51, and the pro-apoptotic BCL-2 family members (PUMA and NOXA). 

This means that when DNA damage occurs, P53 will be stabilized (not degraded by proteolysis) and will become active after phosphorylation and post-translational modifications. 

Figure 3. 

1. It shows that NB/PML can inhibit DNA lesion repair. 

2. Pointing to the reason, the formation of the PML/RAR+P53 complex. 

3.Then P21 causes a G1 phase block. 

How did these three events come about? 

Comments in order of reading: 

Line 40: The mitotic cycle is a series of steps in which chromosomes and other cell material double in order to make two copies. 

Line 110: Nuclear bodies (NBs) 

Line: 357-358 FDA, FDA 

Line 439: "by inducing mutation dependence in BCL-2." An explanation is required as to what the sentence means. 

Line 487: RNI the full name is missing. 

In Tables 1 and 2, there are no references. 

Labeling/titles of the figures

It would be beneficial to change the font style to bold in the case of all the figure legends, e.g. Figure 1. Promyelocytic leukemia is associated with the chromosomal translocation t(15;17)(q22;q21) 67 which results in the fusion of genes PML and RARα. 

Additionally, A figure legend should have a short but descriptive title about the message that it carries, which can be involved in the figure legend below the figure.  

Some of the figures had 2 labels, the title was above, while the legend went below the figure. It should be consistent.

Overall, the quality of the figures is the weakest part of the manuscript.

Line number 108 - there is an unnecessary spacing between the titles and the paragraph 

Line number 205 - there is an unnecessary spacing between the titles and the paragraph 

Line number 252, 254 - there is an unnecessary spacing between the titles and the paragraph 

Line number 506 - there is an unnecessary spacing between the titles and the paragraph 

Line number 517 - there is an unnecessary spacing between the titles and the paragraph 

Line number 529- there is an unnecessary spacing between the first Word: -matory process.  Therefore

Line number 540, 554- there is an unnecessary spacing between the titles and the paragraph 

Line number 583 - there is an unnecessary spacing before Venetoxclax 

Table 4. There are some alignment problems in the table 

Line number 637 - there is an unnecessary spacing between the titles and the paragraph 

Line number 694 - there is an unnecessary spacing between the titles and the paragraph 

Line number 725 - there is an unnecessary spacing between the titles and the paragraph 

Line number 922- there is an unnecessary spacing  

Author Response

Dear Reviewer, 

Manuscript Number: futurepharmacol-2123703

Many thanks for your comments. Certainly, we have considered all comments.

Reviewer 2 Report

The authors performed a complete review of pathogenesis, conventional and possible future treatment of APL with classical t15;17 translocation.

The paper is complete and clear, illustrations appropriate and references updated. 

I only suggest to add a brief section on “alternative”, rare translocation in APL, involving RARA or not, with impact on prognosis and on response to conventional therapy 

Author Response

(The authors gave the same response as above.)
